# Secreted Cytokines within the Urine of AKI Patients Modulate TP53 and SIRT1 Levels in a Human Podocyte Cell Model

**DOI:** 10.3390/ijms24098228

**Published:** 2023-05-04

**Authors:** Lars Erichsen, Chantelle Thimm, Wasco Wruck, Daniela Kaierle, Manon Schless, Laura Huthmann, Thomas Dimski, Detlef Kindgen-Milles, Timo Brandenburger, James Adjaye

**Affiliations:** 1Institute for Stem Cell Research and Regenerative Medicine, Medical Faculty, Heinrich-Heine University Düsseldorf, 40225 Düsseldorf, Germany; 2Department of Anesthesiology, University Hospital Düsseldorf, Heinrich-Heine University Duesseldorf, Moorenstr. 5, 40225 Düsseldorf, Germany; 3Zayed Centre for Research into Rare Diseases in Children (ZCR), EGA Institute for Women’s Health, University College London (UCL), 20 Guilford Street, London WC1N 1DZ, UK

**Keywords:** acute kidney injury, cytokines, TP53, SIRT1

## Abstract

Acute kidney injury (AKI) is a major kidney disease with a poor clinical outcome. It is a common complication, with an incidence of 10–15% of patients admitted to hospital. This rate even increases for patients who are admitted to the intensive care unit, with an incidence of >50%. AKI is characterized by a rapid increase in serum creatinine, decrease in urine output, or both. The associated symptoms include feeling sick or being sick, diarrhoea, dehydration, decreased urine output (although occasionally the urine output remains normal), fluid retention causing swelling in the legs or ankles, shortness of breath, fatigue and nausea. However, sometimes acute kidney injury causes no signs or symptoms and is detected by lab tests. Therefore, the identification of cytokines for the early detection and diagnosis of AKI is highly desirable, as their application might enable the prevention of the progression from AKI to chronic kidney disease (CKD). In this study, we analysed the secretome of the urine of an AKI patient cohort by employing a kidney-biomarker cytokine assay. Based on these results, we suggest ADIPOQ, EGF and SERPIN3A as potential cytokines that might be able to detect AKI as early as 24 h post-surgery. For the later stages, as common cytokines for the detection of AKI in both male and female patients, we suggest VEGF, SERPIN3A, TNFSF12, ANPEP, CXCL1, REN, CLU and PLAU. These cytokines in combination might present a robust strategy for identifying the development of AKI as early as 24 h or 72 h post-surgery. Furthermore, we evaluated the effect of patient and healthy urine on human podocyte cells. We conclude that cytokines abundant in the urine of AKI patients trigger processes that are needed to repair the damaged nephron and activate TP53 and SIRT1 to maintain the balance between proliferation, angiogenesis, and cell cycle arrest.

## 1. Introduction

The understanding of the ethology underlying AKI has fundamentally changed in recent years [1,2]. In the past, research focused on the most severe impairment of the kidney. More recent studies have provided evidence that even minor injuries or impairments of kidney function can lead to severe consequences, such as changes in serum creatinine (sCr) and/or urine secretion (UO) [3]. Today, the term acute kidney injury (AKI) is used for a major kidney disease with a poor clinical outcome. It is characterised as an abrupt (within hours or days) decline in renal function, which includes structural damage to the nephron as well as loss of functionality. It is a common complication in patients admitted to the hospital, with an incidence of 10–15% [4]. This rate even increases for patients who are admitted to the intensive care unit, with the incidence rising to >50% [5]. The classification of AKI includes pre-renal AKI, acute post-renal obstructive nephropathy, and intrinsic acute kidney disease. Of these, only the latter represents a true renal disease, whereas pre- and post-renal AKI are the result of extra-renal disease leading to a decreased glomerular filtration rate (GFR). If these pre- and/or post-renal conditions persist, they eventually develop into cellular damage to the kidney and thus intrinsic renal disease [3]. The current diagnostic approach for AKI is based on an acute decrease in the GFR reflected in an acute increase in the sCr level and/or a decrease in the UO value over a given time interval [3]. According to the current Kidney Disease: Improving Global Outcomes (KDIGO) guidelines, the definition of AKI is classified into three distinct stages: stage 1: creatinine >1.5 times baseline or an increase of ≥0.3 mg/dL within any 48 h period, or a urine volume <0.5 mL/kg for 6–12 h; stage 2: creatinine ≥2.0 times baseline or a urine volume <0.5 mL/kg for ≥12 h; and stage 3: creatinine >3.0 times baseline or increased to ≥4.0 mg/dL or acute dialysis, or a urine volume <0.3 mL/kg for ≥24 h [6]. If these conditions persist for up to seven days, the pathological conditions are referred to as AKI. Between 7 and 90 days, the conditions are referred to as acute kidney disease (AKD), and if they exceed 90 days, chronic kidney disease (CKD) has manifested [7]. In terms of their clinical presentation, kidney diseases are very difficult to identify because, except for urinary tract obstructions, they do not cause any specific signs or symptoms [7]. Furthermore, diagnosis of AKI is hindered by the fact that specific syndromes often co-exist (as reviewed in [7]) or AKI arises as part of other syndromes, such as heart or liver failure or sepsis. In addition, 40% of all AKI cases in the hospital are related to surgical procedures [8], with AKI being associated with cardiac surgery with an incidence ranging from 7 to 40% [9,10,11,12].

Therefore, non-invasive biomarkers for the early detection of AKI are urgently needed. This might improve patient outcomes in the intensive care unit and reduce the risk of progression from AKI to CKD and, ultimately, end-stage kidney disease (ESKD).

Under healthy physiological conditions, kidney cells divide at very low rates [13,14]. In contrast, several studies have reported a dramatic increase in proliferating cells after AKI, which can also be accompanied by transient dedifferentiation [14,15] to repair the injured nephron. This repair process has been identified to be possibly maladaptive, depending on the severity of the injury, involving increased DNA damage and cellular senescence [16]. Therefore, it is not surprising that TP53 and its downstream target, TP21, were found to be upregulated in the kidney after AKI and that inhibition or gene deletion reduces kidney lesions [17,18,19,20,21]. Interestingly, using P53 inhibitors after unilateral ischemia reperfusion injury resulted in reduced fibrosis [20]. Despite this progress in understanding the role of TP53 in the pathology of kidney injury, the underlying mechanism in the progress remains largely unknown. The NAD+-dependent class III histone deacetylase SIRT1 is involved in many biological processes, including DNA damage repair and TP53 activation [22,23]. Furthermore, SIRT1 has been reported to play a protective role in many diseases, including AKI [24,25,26].

In the present study, we screened urine from intensive care patients who developed AKI 24 h and 72 h post-surgery for established kidney biomarkers by employing a kidney-injury cytokine assay. We identified ADIPOQ, EGF and SERPIN3A to be upregulated in urine derived from AKI patients as early as 24 h post-surgery. At 72 h post-surgery, the common cytokines upregulated in the urine samples of both male female patients include VEGF, SERPIN3A, TNFSF12, ANPEP, CXCL1, REN, CLU and PLAU. Furthermore, we applied the urine to our recently published hTERT immortalized podocyte cell line [27] and evaluated the effects of the urine on TP53 and SIRT1 expression.

## 2. Results

### 2.1. Identification of Cytokines Upregulated in Urine Derived from AKI Patients 24 h Post-Operation

Urine samples from six stage 2/3 AKI patients and six patients without AKI were collected 24 h post-surgery at the clinic for anaesthesiology at Heinrich-Heine University Düsseldorf. The patients without AKI were aged between 55 and 75 years—three males and three females. The AKI patients were aged between 34 and 84 years, with the same sex distribution (see Table 1). The urine samples were incubated individually on the human kidney biomarker array, and scanned images are shown in Appendix A. The results are shown in Figure 1. As shown in the dendrogram (Figure 1A), except for P100_M34_AKI3, the samples segregated into two distinct clusters of AKI (red) and healthy controls (blue). The clustering of the AKI group could even further be sub-divided into male and female. Interestingly, the cytokines in the healthy urine seem to be more equally distributed between the genders according to the dendrogram. The heatmap (Figure 1B) and the barplots (Figure 1C,D) show the cytokines found to be significantly regulated as detected by the array.

In the female patients, adiponectin (ADIPOQ), cellular communication network factor 1 (CCN1), epidermal growth factor receptor (EGFR), interleukin 6 (IL6), CC-chemokine ligand 2 (CCL2), thrombospondin 1 (THBS1) and vascular endothelial growth factor (VEGF) were found to be upregulated, while alanyl aminopeptidase (ANPEP), β2-mikroglobulin (B2M), cystatin C (CST3), dipeptidylpeptidase-4 (DDP4), epidermal growth factor (EGF), interleukin 1 receptor antagonist (IL1RN), kallikrein-related peptidase 3 (KLK3), serpin family A member 3 (SERPIN3A), tumour necrosis factor receptor superfamily member 1A (TNFRSF1A), trefoil factor 3 (TFF3) and tumour necrosis factor ligand superfamily member 12 (TNFSF12) were found to be downregulated.

In the male patients, adiponectin (ADIPOQ) and tumour necrosis factor ligand superfamily member 12 (TNFSF12) were found to be upregulated, while epidermal growth factor (EGF), interleukin 10 (IL10), CC-chemokine ligand 2 (CCL2), serpin family A member 3 (SERPIN3A) and tumour necrosis factor alpha (TNFA) were found to be downregulated.

### 2.2. Identification of Cytokines Upregulated in Urine Derived from AKI Patients 72 h Post-Operation

Urine samples from the same six AKI and non-AKI patients were collected 72 h post-surgery. The patient characteristics were similar between the AKI and non-AKI patients (e.g., age, gender, comorbidities; Table 1). The intra-operative data (operation time, cross-clamp time and time on cardiopulmonary bypass) were also comparable. The creatinine levels, as expected, were higher in the AKI group, and so were the lengths of the intensive care and hospital stays. The urine samples were pooled (based on gender and disease state) and incubated on the human kidney biomarker array. The results are shown in Figure 2, while scanned images are shown in Appendix A. As shown in the dendrogram (Figure 2A), the samples segregated into two distinct clusters, with the male AKI samples (black) being closer but distinct from the healthy controls (red) when compared to the female AKI samples (blue). The heatmap (Figure 2B) and the barplots (Figure 2C,D) show the cytokines significantly regulated (*p* < 0.05) as detected by the array. In the female patients, alanyl aminopeptidase (ANPEP), chemokine (C-X-C motif) ligand 1 (CXCL1), interleukin 1 receptor antagonist (IL1RN), renin (REN) and vascular endothelial growth factor (VEGF) were found to be upregulated, while β2-mikroglobulin (B2M), clusterin (CLU), epidermal growth factor (EGF), CC-chemokine ligand 2 (CCL2), matrix metallopeptidase 9 (MMP-9), membrane metalloendopeptidase (MME), kallikrein-related peptidase 3 (KLK3), advanced glycosylation end-product specific receptor (AGER), serpin family A member 3 (SERPIN3A), tumour necrosis factor ligand superfamily member 12 (TNFSF12) and plasminogen activator urokinase (PLAU) were found to be downregulated. In the male patients, alanyl aminopeptidase (ANPEP), epidermal growth factor receptor (EGFR), alpha 2-HS glycoprotein (AHSG), chemokine (C-X-C motif) ligand 1 (CXCL1), CC-chemokine ligand 2 (CCL2), renin (REN) and vascular endothelial growth factor (VEGF) were found to be upregulated, while annexin A5 (ANXA5), clusterin (CLU), cellular communication network factor 1 (CCN1), interleukin 1 receptor antagonist (IL1RN), interleukin 6 (IL6), interleukin 10 (IL10), SKP1-CUL1-F-box protein (SKP1), serpin family A member 3 (SERPIN3A), tumour necrosis factor ligand superfamily member 12 (TNFSF12) and plasminogen activator urokinase (PLAU) were found to be downregulated.

### 2.3. Effect of AKI Stage 2/3 Urine on Podocytes

A number of the chemokines, such as VEGF and EGFR, that were found to be upregulated are capable of either activating or being activated by either TP53 [29,30,31,32] or SIRT1 [33,34]. To test the hypothesis that the chemokines found to be upregulated in the AKI stage 2/3 urine have a direct influence on the expression levels of TP53 and SIRT1, we incubated our recently published immortalized podocyte cell line UM51 Htert [27] with a medium composed of 10% of the patient urine 72 h post-surgery for 120 h. The relative protein expression normalized to GAPDH for TP53 and SIRT1 and the phosphorylation levels of histone 2A (pH2A.X), which is an established biomarker of DNA damage [35], were detected by Western blotting (Figure 3A), and full images of the Western blots are shown in Appendix A. All the proteins were detected at the expected sizes of 110 kDa for SIRT1, 15 kDa for pH2A.X, 53 kDa for p53 and 38 kDa for GAPDH. While the SIRT1 expression was not detectable in the podocytes under control conditions, it significantly increased in the podocytes incubated with healthy urine to 600% (*p* < 0.01) and was even found to be further elevated when the podocytes were incubated with AKI stage 2/3 urine to 1051% (*p* < 0.01). In contrast, the TP53 levels were found to be significantly downregulated by 85% when the cells were incubated with healthy urine (*p* > 0.05). When the cells were incubated with the AKI stage 2/3 urine, the TP53 expression was found to be significantly upregulated to 150% (*p* < 0.05). Accordingly, the phosphorylation levels of H2A.X were found to be upregulated under both conditions, with healthy urine increasing the phosphorylation levels by 280% and the AKI 2/3 urine by 700%. Both changes were found to be significant (*p* < 0.05). The qRT-PCR analysis of TP53 and SIRT1 (Figure 3B) revealed that the mRNA expression of both was upregulated in the podocytes treated with human urine. Healthy urine induced a 2.5-fold increase in the TP53 mRNA expression and a 1.7-fold increase in the TP51 expression, while AKI stage 2/3 urine induced an upregulation of the TP53 mRNA by 3.5-fold and the SIRT1 mRNA by 4.2-fold. Finally, we applied immunofluorescence-based detection of TP53 to the cells exposed to AKI 2/3 urine (Figure 3C), revealing an increase in positively stained cells.

## 3. Discussion

In this study, by comparing the urine-secreted cytokines of non-AKI and AKI 2/3-patients after cardiac surgery, we identified a panel of cytokine specifically upregulated in the urine derived from AKI patients. Since kidney diseases can be difficult to identify in the clinic because they do not cause any specific signs or symptoms [7], and as AKI can arise as part of other syndromes such as heart or liver failure and sepsis, the variability between individuals and genders has to be accounted for. Since we only had access to a limited sample pool for this study, we aimed at excluding individual variability by pooling the samples. Gender-specific analyses were carried out to avoid variations due to the influence of sex hormones, such as testosterone and oestradiol, in the modulation of SIRT1 expression [36,37,38]. While the male urine samples 24 h post-surgery revealed only seven cytokines as significantly altered, the female urine samples contained a plethora of significantly altered cytokines. The common cytokines in both the female and male urine samples provided 24 h post-surgery were the upregulation of adiponectin (ADIPOQ) and the downregulation of epidermal growth factor (EGF) and serpin family A member 3 (SERPIN3A). In contrast, the analysis of the urine from the stage 2/3 AKI patients 72 h post-surgery showed a dramatic increase in the regulated cytokines. Here, the male and female samples showed the upregulation of vascular endothelial growth factor (VEGF), alanyl aminopeptidase (ANPEP), chemokine (C-X-C motif), ligand 1 (CXCL1) and renin (REN), while clusterin (CLU), SERPIN3A, plasminogen activator urokinase (PLAU) and tumour necrosis factor ligand superfamily member 12 (TNFSF12) were found to be downregulated. Of note, the upregulation of ADIPOQ detected in the 24 post-surgery urine was reduced after 72 h. The same trend was observed for EGF in the female urine, while the downregulation was persistent in the male AKI urine samples 72 h post-surgery. Interestingly, the downregulation of secreted interleukin 10 was only observed in the male AKI urine samples 24 h and 72 h post-surgery. Based on these results, we suggest ADIPOQ, EGF and SERPIN3A as potential cytokines that might be able to detect AKI as early as 24 h post-surgery. For the later stages, as common cytokines for the detection of AKI in both male and female patients, we propose VEGF, SERPIN3A, TNFSF12, ANPEP, CXCL1, REN, CLU and PLAU. These markers in combination might present a robust strategy for diagnosing AKI as early as 24 h or 72 h post-surgery. Many of these cytokines have already been described in the context of AKI [39,40,41]. The full list of significantly regulated cytokines is given in Table 2.

At the molecular level, ADIPOQ might be of special interest, since it has been described to directly regulate the expression of three of the identified cytokines that were upregulated 72 h post-surgery, namely CXCL1, VEGF and RENIN.

In cancer cells, adiponectin has been shown to induce CXCL1 secretion and thereby promote tumour angiogenesis [29]. In addition, CXCL1 has been reported to stimulate angiogenesis via the VEGF pathway [42,43], and it has even been suggested as a biomarker for patients suffering from colon cancer [44]. Furthermore, it has been shown that inhibition of the renin–angiotensin system results in significantly increased adiponectin levels [45]. Therefore, we propose that the observed upregulation of secreted RENIN in the AKI urine samples 72 h post-surgery might be causative for the downregulation of secreted ADIPOQ, while the early upregulation of secreted ADIPOQ results in the activation of the angiogenesis-related genes CXCL1 and VEGF, thus reflecting the initiation of the repair processes in the damaged nephron. For instance, VEGF has been shown to be an important mediator of the early and late phase of renal protective action after AKI in the context of stem cell treatment [40].

EGF and EGFR both have been associated with AKI, and low levels of EGF have been reported to be predictive of AKI in newborns [46,47]. Of note, EGF has been reported to induce several matrix metalloproteinases [48] and, therefore, the identified downregulation of EGF might be causative for the also observed downregulation of MME and MMP9 in the female AKI urine 72 h post-surgery. EGFR has been reported to play an important role in renal recovery from AKI through PI3K-AKT-dependent YAP activation [49]. Indeed, the EGFR promoter bears several p53-responsive elements [50], and it has been reported that renal fibrosis is associated with EGFR and p53 activation [51]. Interestingly, the FDA approved test “NephroCheck^®^”, which is in clinical use and predicts the risk of developing moderate to severe AKI within 24 h [52], uses the following two biomarkers: tissue inhibitor of metalloproteinase 2 (TIMP-2) and insulin-like growth factor binding protein 7 (IGFBP7). Both are secreted by renal tubular cells in response to cellular stress and are involved in G1-phase cell-cycle arrest [53]. This is achieved by the transcriptional activation of p27 via TIMP-2 and p53 and p21 via IGFBP7 [54,55], further highlighting the involvement of TP53 activation in the clinical manifestation of AKI. Another study carried out by Montero et al. also documented that VEGF and p53 expression levels are significantly correlated and can be used as independent prognostic factors [30]. Our results suggest that secreted EGFR and VEGF are sufficient for the upregulation of p53 mRNA and protein levels in our immortalized podocyte cell line, although further experiments need to be conducted to verify this hypothesis. Furthermore, another study carried out by Pierzchalski et al. showed that p53 is able to induce apoptosis via the activation of the renin–angiotensin system [31]. In two recent studies carried out in our laboratory, we demonstrated the devastating effect of the activated renin–angiotensin system on podocyte morphology and functionality [27,56]. Therefore, we propose that the observed upregulated secretion of EGFR and VEGF mainly triggers angiogenesis and the activation of p53 and the renin–angiotensin system. This mode of action might be a plausible explanation for the progression from AKI to CKD. Considering the constant secretion of these factors, they should be investigated in urine samples from patients with AKI for more than three days after surgery and in patients who progress to CKD. Interestingly, our recently published study investigating the secretome of urine from a CKD patient cohort from Ghana revealed the downregulation of secreted EGFR and VEGF [57]. This makes it tempting to speculate that EGFR and VEGF might also be involved as initiators of AKI and downregulated in the progression to CKD. In particular, VEGF, in the context of CKD, has been shown to have therapeutic potential by inducing renal recovery [58]. The exact mechanism underlying this protective effect of VEGF is unknown, although kidney tissue after AKI is characterized by impaired endothelial proliferation and mesenchymal transition—both contribute to vascular refraction and, subsequently, to the progression to CKD [59]. So far, the clinical diagnosis and monitoring of AKI is based on the clearance of urea and creatinine [2]. This means that an impairment of the filtration barrier of the nephron already has happened, and the repair process of the kidney is monitored. A clear advantage of a cytokine dosage-based approach is the possible detection of the development of AKI prior to or at the onset of reduced kidney function. This might be especially true since we found an accumulation of ADIPOQ in the urine of patients who developed AKI as early as 24 h post-surgery. This factor is known to upregulate cytokines such as VEGF, CXCL1 and REN in a feed-forward cycle and, therefore, activate distinct pathways that are known to contribute towards the development and progression of AKI. By detecting the accumulation of urine-secreted cytokines such as ADIPQ as early as possible, therapeutic approaches to prevent the breakdown of the filtration barrier might be initiated, with great benefits for the patients. Nevertheless, this needs to be evaluated in further studies. Another great benefit of a cytokine-based approach is that the test can be conducted in a non-invasive, fast and cheap manner by simply collecting the urine and detecting the aberrant regulated cytokines on a suitable device, such as a flow cell.

Finally, we found an upregulation of SIRT1 in the podocytes incubated with the patient-derived urine, which was found to be even more pronounced when AKI stage 2/3 urine was used. SIRT1 has been reported to be renal-protective during AKI [24,60]. This protective effect might be caused by two distinct pathways that are regulated by SIRT1 activity. First, p53 is a direct substrate of the deacetylation activity of SIRT1 by regulating p53 activity, since p53 acetylation promotes the transactivation of numerous genes regulating cell-cycle arrest, apoptosis and metabolic targets [61]. In this context, it has been shown that SIRT1 overexpression promotes cellular survival in the presence of a cellular stressor [62,63]. The second pathway directly influenced by SIRT1 is the renin–angiotensin pathway [34]. In this context, the overexpression of SIRT1 has been reported to attenuate angiotensin II-induced vascular remodelling and hypertension [64], which might be caused by SIRT1-induced downregulation of the angiotensin II type 1 receptor (AGTR1) [65]. SIRT1 has been reported to increase the lifespan in lower organisms, such as *Caenorhabditis elegans* [66] and *Drosophila melanogaster* [67], and recently, we reported the age-associated decline in SIRT1 mRNA and protein in urine-derived renal progenitor cells [68]. Since AKI has been described as a condition of renal senescence [69], the expression level of SIRT1 seems to be of major importance for the prevention of AKI. Nevertheless, this study is limited by the small sample size. This is especially true with regard to the complex nature of human urine, which contains a plethora of cytokines and molecules harbouring possible mechanistic influences on the observed results. Therefore, further experiments in which the cells are exposed to recombinant EGFR or VEGF should be conducted, although that is beyond the limits of the current study.

In summary, we propose putative urine-based cytokines (24 and 72 h post-surgery) for the detection of AKI, which are male-enriched (AHSG), female-enriched (CCN1, IL6, CCL2, THBS1, IL1RN) and common in both (VEGF, SERPIN3A, TNFSF12, ANPEP, CXCL1, REN, CLU and PLAU). Furthermore, mechanistically, the cytokines ADIPOQ, EGF, EGFR, REN and VEGF present in the urine of AKI stage 2/3 patients trigger processes such as angiogenesis that are needed to repair the damaged nephron and the activation of p53 and SIRT1 to maintain the balance between proliferation, angiogenesis, repair and cell-cycle arrest. On this basis, we propose a specific signalling cascade induced by damage to the kidney, resulting in the establishment and propagation of AKI (Figure 4).

## 4. Materials and Methods

### 4.1. Study Design

After approval by the local ethics committee of Heinrich-Heine University Duesseldorf, Germany (local trial number 5803, clinicaltrials.gov: NCT03089242), adult patients scheduled for cardiac surgery were enrolled after they had provided written informed consent to participate in the study.

Urinary samples from 6 patients with moderate to severe AKI (AKI stage 2 or 3) and from 6 matched controls (heart surgery, no AKI post-operatively) were centrifuged (3500 rpm, 5 min) and stored until further analysis.

### 4.2. Secretome Analyses

The urine samples were analysed using the Proteome Profiler Human Kidney Biomarker Array Kit (#ARY019) distributed by Research and Diagnostic Systems, Inc. (Minneapolis, MN, USA), as described by the manufacturer and in our previous publication [57]. In brief, the membranes were blocked with the provided blocking buffer and each urine sample was incubated with 15 μL of reconstituted Detection Antibody Cocktail for 1 h at room temperature. After removing the blocking buffer, the urine samples were incubated on the membranes at 4 °C overnight on a rocking platform. After three consecutive washing steps, each for 10 min at room temperature on a rocking platform, the membranes were incubated with the diluted Streptavidin–HRP antibody mix. After three consecutive washing steps, each for 10 min at room temperature on a rocking platform, the fluorescence signals were visualized with enhanced luminescence (WesternBright Quantum, Advansta, San Jose, CA, USA).

The obtained images were analyzed using Image J software [70] with the Microarray Profile plugin by Bob Dougherty and Wayne Rasband (https://www.optinav.info/MicroArray_Profile.html accessed on 10 March 2023). The integrated density generated by the Microarray Profile plugin function Measure RT was used for the follow-up processing, which was performed in the R/Bioconductor environment [71]. The arrays were normalized by employing the Robust Spline Normalization from the Bioconductor lumi package [72]. The threshold for background intensities was defined at 5% of the range between the maximum and minimum intensity, and the detection *p*-value was calculated according to the method described in Graffmann et al. [73]. Cytokines detected in the AKI and control condition with a *p*-value from the R package limma of less than 0.05 controlled for the false discovery rate (FDR < 0.25) and with a ratio greater than 1.2 (inverse ratio < 0.8333 for downregulation) were considered upregulated [28].

### 4.3. Cell Culture Conditions

The cell line UM51-hTERT was derived and cultured as described in ref. [27]. For the experiments, the non-AKI and AKI 2/3 patient urine samples obtained 72 h post-surgery were pooled and mixed with the culture medium to a final concentration of 10%. This concentration was observed to be acceptable for cell culture experiments in terms of a lack of contamination and cell death. The cells were incubated with this medium for five days.

### 4.4. Relative Quantification of Podocyte-Associated Gene Expression by Real-Time PCR

The real-time PCR of the podocyte-associated gene expression was performed as follows. Real-time measurements were carried out on the Step One Plus Real-Time PCR Systems using a MicroAmp Fast Optical 384 Well Reaction Plate and Power Sybr Green PCR Master Mix (Applied Biosystems, Foster City, CA, USA). The amplification conditions were denaturation at 95 °C for 13 min, followed by 37 cycles of 95 °C for 50 s, 60 °C for 45 s and 72 °C for 30 s. The primer sequences are listed in Appendix A.

### 4.5. Immunofluorescence-Based Detection of Protein Expression

The cells were fixed with 4% paraformaldehyde (PFA) (Polysciences, WA, USA). The unspecific binding sites were blocked by incubation with blocking buffer containing 10% normal goat or donkey serum, 1% BSA, 0.5% triton, and 0.05% tween for 2 h at room temperature. Incubation of the primary antibody was performed at 4 °C overnight in staining buffer (blocking buffer diluted 1:1 with PBS). After at least 16 h of incubation, the cells were washed three times with PBS/0.05% tween and incubated with a 1:500 dilution of secondary antibodies. After three additional washing steps with PBS/0.05% tween, the cell and nuclei were stained with Hoechst 1:5000 (Thermo Fisher Scientific, Waltham, MA, USA). Images were captured using a fluorescence microscope (LSM700; Zeiss, Oberkochen, Germany) with Zenblue software (Zeiss). Individual channel images were processed and merged with Fiji. Detailed information on the used antibodies is given in Appendix A.

### 4.6. Western Blotting

Protein isolation was performed via lysis of the cells in RIPA buffer (Sigma-Aldrich, St. Louis, MO, USA) supplemented with complete protease and phosphatase inhibitors (Roche, Basel, Switzerland). The proteins were separated on a 7.5% Bis-Tris gel and blotted onto a 0.45 µm nitrocellulose membrane (GE Healthcare Life Sciences, Chalfont St. Giles, UK). After blocking the membranes with 5% skimmed milk in Tris-buffered saline tween (TBS-T), they were incubated overnight with the respective primary antibodies (Appendix A). The membranes were washed 3× for 10 min with TBS-T. Secondary antibody incubation was performed for 1 h at RT and the membranes were subsequently washed 3× for 10 min with TBS-T. Amersham ECL Prime Western Blotting Detection Reagent was used for the chemiluminescent detection (GE Healthcare Life Sciences) and captured with the imaging device Fusion FX.

### 4.7. Statistics

The data are presented as arithmetic means + standard error. In total, three independent experiments were performed and used for the calculation of the mean values. Statistical significance was calculated using the two-sample Student’s *t*-test or with Mann–Whitney U test to compare the variables between the two patient groups. The association between the categorical variables and the two groups (no AKI vs. AKI patients) was tested with the Chi-squared test. For all the tests, *p*-values < 0.05 were considered significant.

## Figures and Tables

**Figure 1 ijms-24-08228-f001:**
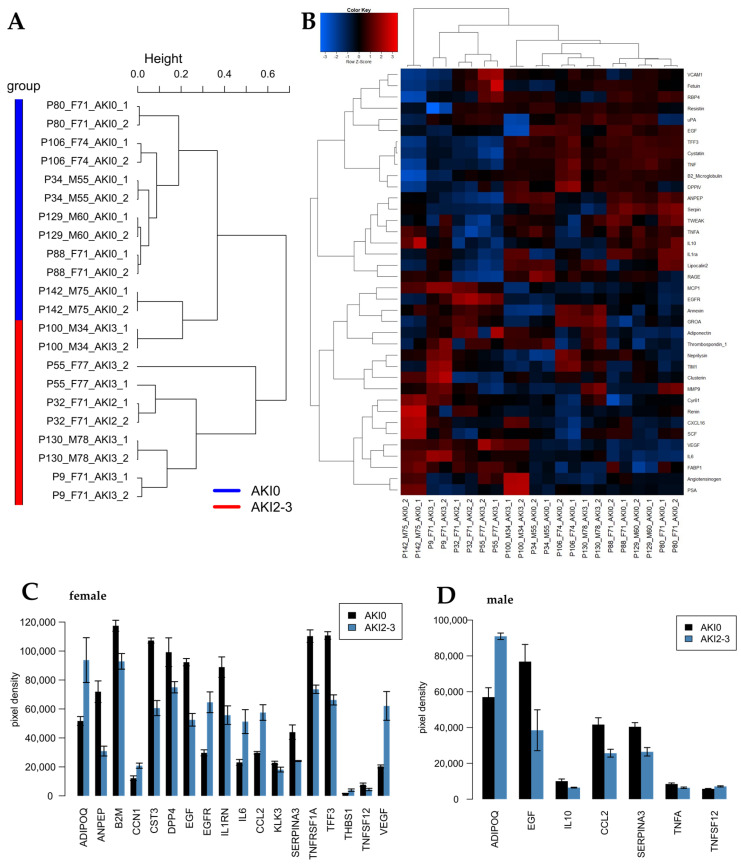
AKI cytokines identified in patients 24 h post-surgery. (**A**) Experiments clustered into CKD (*n* = 6) and healthy control (*n* = 6) based on the global kidney cytokine expression. (**B**) Heatmap and (**C**) barplot of markers in females and barplot of markers in males (**D**). The histograms represent means and the error bars standard errors of the mean (*n* = 3). The cytokines depicted here were differentially upregulated with a *p*-value (test of the R package limma (Smyth 2004 [28])) *p* < 0.05 and a ratio > 1.2, or downregulated with a *p* < 0.05, FDR < 0.25 and ratio < 0.8333.

**Figure 2 ijms-24-08228-f002:**
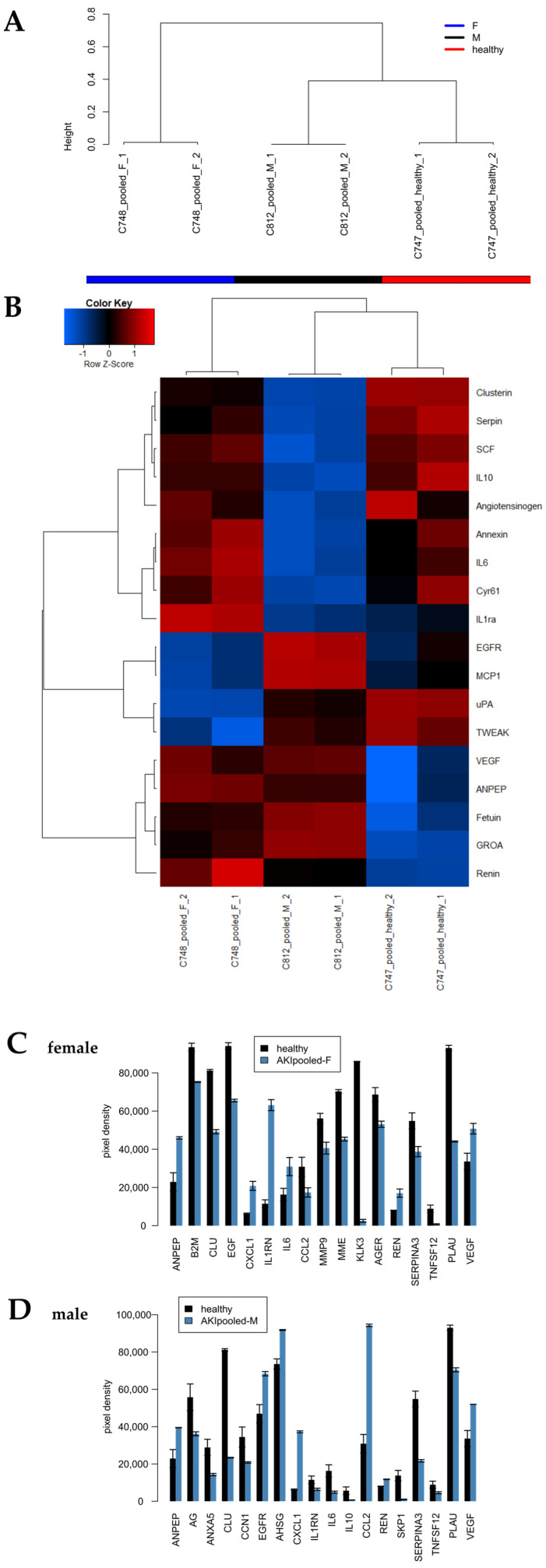
AKI cytokines identified in patients 72 h post-surgery. (**A**) Experiments clustered into CKD (*n* = 6) and healthy control (*n* = 6) based on the global kidney cytokine expression. (**B**) Heatmap and (**C**) barplot of markers in females and barplot of markers in males (**D**). The histograms represent means and the error bars standard errors of the mean of technical duplicates (*n* = 2). The cytokines depicted here were differentially upregulated with a *p*-value (test of the R package limma) *p* < 0.05 and a ratio > 1.2, or downregulated with a *p* < 0.05, FDR < 0.25 and ratio < 0.8333.

**Figure 3 ijms-24-08228-f003:**
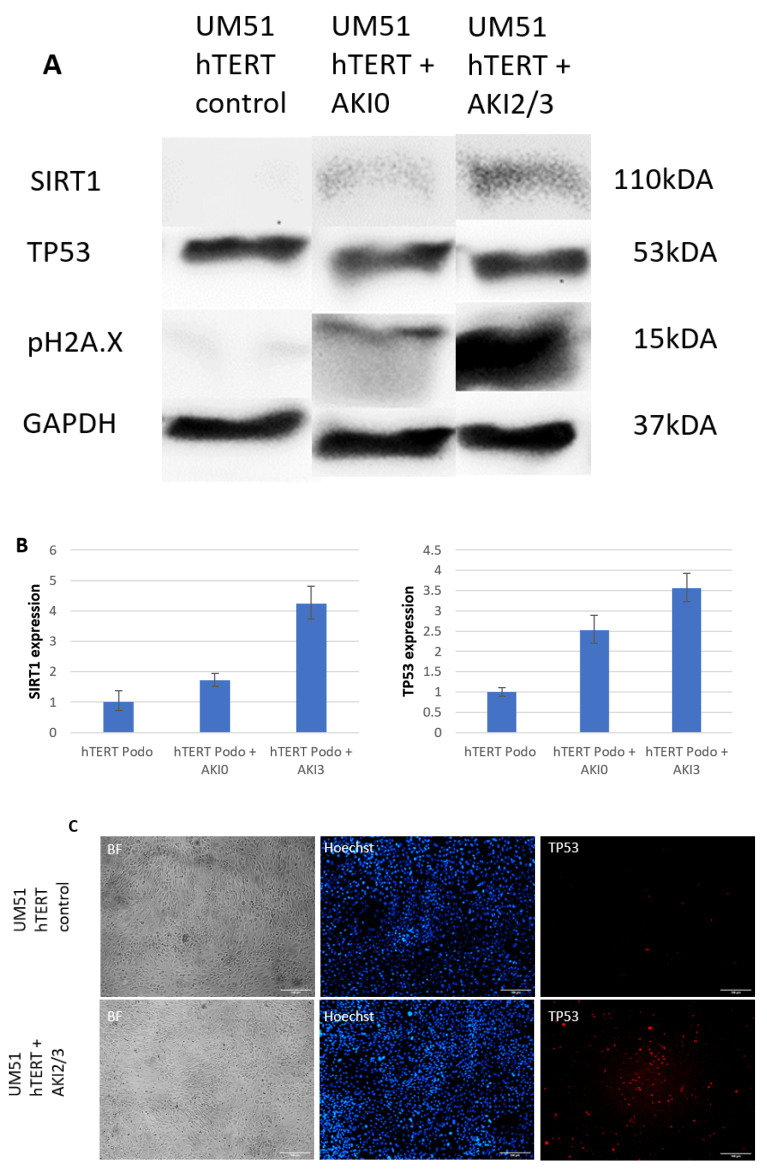
Upregulation of TP53 and SIRT1 induced by AKI stage 2/3 urine in podocytes. The podocytes were incubated with 72 h post-surgery healthy (*n* = 6) and AKI stage 2/3 urine (*n* = 6) for 5 days. The relative protein expression normalized to GAPDH for SIRT1, TP53 and H2A.X phosphorylation was detected by Western blot (**A**). The mRNA expression of TP53 and SIRT1 was determined by quantitative real-time PCR (**B**). The TP53 expression in the podocytes exposed to AKI 2/3 urine was detected by immunofluorescence-based staining (**C**). Scale bars indicate 100 µm.

**Figure 4 ijms-24-08228-f004:**
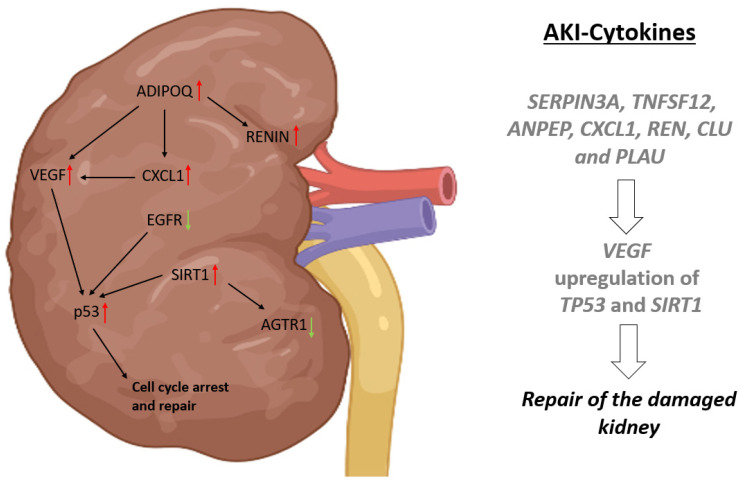
Proposed AKI signalling cascade. Urine-containing cytokines for the detection of AKI, which are male-enriched (AHSG), female-enriched (CCN1, IL6, CCL2, THBS1, IL1RN) and common in both (VEGF, SERPIN3A, TNFSF12, ANPEP, CXCL1, REN, CLU and PLAU). Furthermore, mechanistically, the cytokines ADIPOQ, EGF, EGFR, REN and VEGF present in the urine of AKI 2/3 patients trigger processes such as angiogenesis that are needed to repair the damaged nephron and the activation of TP53 and SIRT1 to maintain the balance between proliferation, angiogenesis, repair and cell-cycle arrest. Upregulated Cyokines and Proteins are indicated by a red arrow, while downregulated ones are indicated by a green arrow. Created with BioRender.com.

**Table 1 ijms-24-08228-t001:** Patient cohort.

Parameter	No AKI (*n* = 6)	AKI 2 and 3 (*n* = 6)	*p*-Value
Age [years]	68 ± 8	69 ± 18	0.37
Gender			
Female [%]	3 (50)	3 (50)	1.0
Male [%]	3 (50)	3 (50)	1.0
Comorbidities			
Obesity [%]	2 (33)	3 (50)	1.0
Hypertension [%]	5 (93)	4 (67)	1.0
Diabetes Mellitus [%]	2 (33)	2 (33)	1.0
CKD [%]	0 (0)	2 (33)	0.455
Preoperative creatinine [mg/dL]	1.18 ± 0.44	1.85 ± 0.90	0.126
Operation time [minutes]	282 ± 66	287 ± 86	0.699
CPB time [minutes]	191 ± 65	170 ± 51	0.589
X-clamp [minutes]	109 ± 33	103 ± 52	1.0
Creatinine postop. [mg/dL]	1.2 ± 0.34	1.43 ± 0.42	0.268
Creatinine on day 3 [mg/dL]	1.08 ± 0.50	2.03 ± 1.29	0.097
Lactate postop. [mmol/L]	2.5 ± 0.9	7.5 ± 3.2	**0.004**
Lactate on day 3 [mmol/L]	2.2 ± 2.5	1.7 ± 0.5	0.296
APACHE-II postop.	26 ± 2	32 ± 3	**0.008**
Ventilation [hours]	20 ± 8	186 ± 253	**0.020**
LOICUS [days]	2 ± 2	13 ± 11	**0.014**
LOHS [days]	18 ± 12	42 ± 20	**0.030**
Last creatinine [mg/dL]	1.03 ± 0.33	2.07 ± 0.72	**0.016**
Mortality in hospital	0 (0)	1 (17)	1.0

Matched no AKI vs. AKI stage 2 and 3 data are expressed as the mean ± standard deviation or number (percentage). A statistical analysis was performed with the Mann–Whitney U test to compare the variables between the two groups; the association between the categorical variables and the two groups (no AKI vs. AKI patients) was tested with the Chi-squared test. For all the tests, *p*-values < 0.05 were considered significant. LOICUS = length of intensive care unit stay after surgery; LOHS = length of hospital stay; X-clamp = aortic cross-clamping.

**Table 2 ijms-24-08228-t002:** Summary of the identified cytokines for the early diagnosis of AKI.

Cytokine	Female 24 h Post-Surgery	Male 24 h Post-Surgery	Female 72 h Post-Surgery	Male 72 h Post-Surgery
ADIPOQ	Up	Up		
CCN1	Up			Down
EGFR	Up			Up
IL6	Up			
CCL2	Up	Down	Down	Up
THBS1	Up			
VEGF	Up		Up	Up
ANPEP	Down			
B2M	Down		Down	
CST3	Down			
DDP4	Down			
EGF	Down	Down	Down	
IL1RN	Down		Up	Down
KLK3	Down			
SERPIN3A	Down	Down	Down	Down
TNFRSF1A	Down			
TTF3	Down			
TNFSF12	Down	Up	Down	Down
IL10		Down		Down
TNFA		Down		
ANPEP			Up	Up
CXCL1			Up	Up
REN			Up	Up
CLU			Down	Down
MMP9			Down	
MME			Down	
KLK3			Down	
AGER			Down	
PLAU			Down	Down
AHSG				Up
ANXA5				Down
IL6				Down
SKP1				Down

## Data Availability

All data is available.

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
