# Peer review of "Secreted Cytokines within the Urine of AKI Patients Modulate TP53 and SIRT1 Levels in a Human Podocyte Cell Model"

_ijms, 2023, doi:10.3390/ijms24098228_

Round 1

Reviewer 1 Report

This is a straight forward study that is properly executed. It consists of urine analysis of samples obtained from a small group of small group of patients. The results are clear. The sample size is small so some statistics would be helpful.

Figures 1C,D, 2C,D and 3B are bar graphs showing standard deviation or standard error bars. Please note which it is.

Also, statistics could be done to show significance.There is no statistics done or described in the methods section.

It is mentioned at the beginning of the results that there were 6 AKI and 6 healthy patients but it is worth verifying in the figure legends of Figure 1,2,3 the sample size used to generate the graphs.

Minor grammatical/editing mistakes. Please correct.

Reviewer 2 Report

In this article, the authors analyse the variation in the expression of some urinary cytokines in AKI patients at stage 2/3 versus patients who do not develop AKI 24 hours and 72 hours post-surgery, highlighting the presence of some variations, which are proposed as possible even short-term (24-hour) marker of the onset of AKI. The argument is certainly interesting, but the results shown are unfortunately negatively influenced above all by the small number of patients (subdivided further into 2 groups of three females and three males) analysed.

in a second part the authors evaluate the changes induced on p53 and sirt1 in a podocytes cell line they developed, following exposure to medium containing 10% urine of AKI0 patients versus AKI2/3 patients.

Some minor and major revisions are needed:

1)      Introduction:

Very well written, it provides a clear overview of the problem and the final aims of the study.

However in Line 36, I suggest the authors cite a more recent comprehensive review by Kellum et al 2021 https://doi.org/10.1038/s41581-021-00410-w which refers to new ai Conceptual advances and evolving terminology in acute kidney disease

2)      Result section:

As initially said, I believe that one of the major limitations of the study is in the rather small number of patients analyzed (6) who were further divided into two groups consisting of only 3 males and 3 females. This limitation should be clearly underlined in the discussion section.

Paragraph 2.1

Figure 1A

a)       in the dendogram (panel A) it is evident that the urine samples of one of the AKI patients  (P100_M34_AKI3_1 and 2) cluster with those of the non-AKI patients, healthy patients, before than with the other AKI patients. The statement "the samples segregated into two distinct clusters of AKI (red) and healthy controls 100(blue)" on line 100 should be changed accordingly (see figure 1 panel A pdf attached).

b)      b) in figure 1 panels C and D the writing male and female has probably been reversed with respect to what is written in the figure legend

c)       in figure 1 panels D and C What do the histograms represent: means and sem or mean and standard deviation? What is the number of data? Was a statistical analysis done to evaluate the variation between aki 0 and aki 2/3? If so, which one? Could it be plotted on the graph (* for p<0.05 and ** for p<0.01, for example)? Or, if for all there is a statistically significant variation, could it be reported in the legend of the figure?

On page 4 the author illustrates the results of the experiment conducted with urine from patients 24 hours post-intervention: they state that some proteins are upregulated and others downregulated: is the reported variation statistically significant? Was a statistical test conducted to evaluate whether and which of the variations observed are statistically significant? What is the number of data?

Paragraph 2.2

It is not clear to me and it is not explained why the authors analysed the urine collected at 24 hours individually and instead created a pool of urine from the different patients to conduct the analysis at 72 hours? In this way they reduce the possible variability of analyses between different subjects (remember that one clusters with the healthy at 24 hours), risking masking small but significant variations, and obtaining a large variation as positive which is however caused by a large variation in a only individual. What was the reason why this experimental choice was made?

the same authors rightly state in discussion (line 202) “variability between in individual and gender has to be accounted fo”r, and this makes the reason for this choice even less clear

Figure 2 panel C and D

a)       in figure 1 panels C and D the writing male and female has probably been inverted. Also the vertical axis label overlaps with the numeric values, making the graph difficult to read and should be moved.

b)      in figure 1 panels D and C What do the histograms represent: means and sem or mean and standard deviation? What is the number of data? Was a statistical analysis done to evaluate the variation between aki 0 and aki 2/3? If so, which one? Could it be plotted on the graph (* for p<0.05 and ** for p<0.01, for example)? Or, if for all there is a statistically significant variation, could it be reported in the legend of the figure?

in the presentation of the original images, could the author write the patient code (or sample mix in the case of the 72 hour sample) under each plot? this way the presentation would be more useful

Paragraph 2.3

Why did the authors choose to expose the cells to 10% urine medium? on what hypothesis did they choose this concentration? They should write it here or at least in the materials and methods section.

Were the cells exposed x 5 days to urine medium with no change of medium? How did the viability of the cells vary at the end of the 5th day? Was it evaluated?

Figure 3. I suggest moving the quantification of the western blot to figure 3, to clarify the statements made in the text of the article.

By the way, in the western blot quantification graphs, what do the bars represent? mean? why hasn't the sem or the standard deviation been calculated and reported in order to be able to evaluate whether the variation is significant? How many experimental repetitions were done for the western blot? Could they indicate it in the legend?

Page 8 line 174 p=0.01. Do the authors mean p<0.01? How was the statistical test done? what is the number of data?

Page 8 line 175 p=0.01. Do the authors mean p<0.01? How was the statistical test done? what is the number of data?

Page 8 line 177 p>0.05. if p>0.05 then there is no significant change contrary to what the authors claimed. How was the statistical test done? what is the number of data?

Page 8 line 181 p<0.05. How was the statistical test done? what is the number of data?

Figure 3 panel B How was the statistical test done? what is the number of data?

Figure 3 panel C

Figura 3 the scalebar is missing, at what magnification were the images acquired?

Discussion

a)       Pag 9 Line 202: the authors state (line 202) variability between in individual and gender has to be accounted for, they should explain here why they created a 72-hour urine pool of different patients

b)      Pag 10 line 241: Misspelling: ow should probably be replaced with low

c)       Pag 10 line line 257: the authors state "From our results, we conclude that secreted EGFR and VEGF are sufficient for upregulating p53 mRNA and protein levels in our immortalized podocyte cell line."

However, the authors make this affermation by exposing the cells to urine, a complex mixture in which EGFR and VEGF are ALSO impaired in patients, as well as many other substances including for example urea and uremic toxins (see doi: 10.1111/j.1525-139X.2009.00598.x. as an example) whose effects can add to those of EGFR. and VEGF or all other cytokines, generating a big confusion.

The authors should conduct experiments with EGFR and/or VEGF alone in their experimental model before they can draw a conclusion that seems so definitive, or they should replace concluding with “data suggest that…but further experiments need to be conducted to verify this hypothesis”.

It would be appropriate for the authors to explicitly state that the proposed mechanism, albeit coherent, is based on an experimental observation and on a subsequent series of purely speculative statements based on information drawn from the literature. Although correct, further experiments will need to be conducted to demonstrate the proposed mechanism

d)      The markers that are currently used in the clinic to evaluate AKI are only the clearance of urea and creatinine, which is routinely quantified and monitored in patients throughout the course of AKI ( https://doi.org/10.1038/s41581-021- 00410-w), the authors should better highlight what could be the advantages of a cytokine dosage compared to the classic one of urea and creatinine

e)      The authors should highlight and discuss the limitations of this experimental work: limited number of subjects, cells are exposed to urine and not to single cytokines, other molecules present in this complex solution could influence the observed results. The cytokine variations were obtained from urine samples of people who underwent cardiac surgery (and therefore probably had heart problems that could have already created kidney damage): the authors think that these variations can be extended and evaluated also in patients with other pathologies?

Materials and methods

A paragraph on the statistical analysis of the data is missing: what test did they use to evaluate if the variations observed were statistically significant?

Table 1 Is it possible to have a statistical analysis of the variations observed between patients? what are the characteristics of the two cohorts of Aki and non-Aki patients different from each other? It should be highlighted in the table

Section 4.3 Were the cells grown in medium for 5 consecutive days without any change of medium? Was viability assessed at the end of this period?

Section 4.5

Did the authors conduct immunofluorescence experiments in which cells were exposed to only the secondary antibody as a control? Was autofluorescence evaluated? It should be listed in this section

Round 2

Reviewer 2 Report

I thank the authors for their thorough review and for their punctual comments. After the changes made to the article, I think it has improved overall.

I suggest the authors include a scalebar in the immunofluorescence images, which shouldn't be too complicated. Immunofluorescence controls are missing, but given the presence of a western blot, they may be of minor importance in this case. Statistical analyzes were included and some of the limitations of the work were discussed. For this reason I think the article can be published without further changes.